# Sports Injury Surveillance Systems: A Scoping Review of Practice and Methodologies

**DOI:** 10.3390/jfmk9040177

**Published:** 2024-09-26

**Authors:** Damien Costello, Ed Daly, Lisa Ryan

**Affiliations:** 1Department of Sport, Exercise and Nutrition, School of Science and Computing, Atlantic Technological University, H91 T8NW Galway, Ireland; damien.costello@atu.ie (D.C.); ed.daly@atu.ie (E.D.); 2Irish Concussion Research Centre (ICRC), Atlantic Technological University, H91 T8NW Galway, Ireland

**Keywords:** injury surveillance, mTBI, concussion, sport, multi-sport, injury management

## Abstract

**Background:** Injury prevention/reduction strategies are driven by data collected through injury surveillance systems. The aim of this review was to describe injury surveillance systems that are used for ongoing surveillance in either a professional or amateur sporting environment. This was an update to a review done in 2015 to determine the gaps in injury surveillance. **Methods:** A systematic search process of five databases (MEDLINE, CINAHL, EMBASE, SCOPUS and ProQuest) was used to discover published research that presented methodological data about the injury surveillance systems implemented by clubs and organisations for ongoing surveillance. Inclusion criteria centred on the population under surveillance and the ongoing nature of that activity. Data extracted and summarised included the level of athlete under surveillance, the data collection mechanism and the personnel involved, the injury definitions applied and the date and country of origin to provide a comprehensive picture of the systems. **Results:** A total of 21 systems were documented as being used in ongoing injury surveillance, with 57% of these exclusively in the professional/elite landscapes and 33% at the amateur level. Surveillance systems cater for one sport per athlete entry so there is a gap in research for multi-sport athletes at the amateur level, especially where there is no early specialisation in a multi-sport participation environment. **Conclusions:** Research in this area will lead to a better understanding of subsequent injury risk for multi-sport athletes who have a higher athlete exposure than single-sport athletes.

## 1. Background

Carefully designed injury surveillance programmes, along with accurate data capture and careful analysis of data, are essential components for sports injury and illness prevention programmes [1]. Surveillance is the ongoing, systematic collection, analysis and interpretation of health data. These data are essential to the planning, implementation and evaluation of health practices, which are closely integrated with the dissemination of these data [2]. Injury surveillance is a part of the sequence of prevention identified by Van Mechelen in 1997. It is necessary to identify the sources and causes of injury to support the design of an injury prevention programme. Injury prevention programmes are informed through data collected by injury surveillance systems, with studies in various sports including rugby [3] and ice hockey [4] supporting an injury rate reduction with appropriate interventions. However, the use of injury frequency data alone to assess the relative risk of injury across different sports leads to erroneous conclusions [5]. Injury prevention programmes are also informed by other factors such as participation frequency, level (amateur, semi-professional or professional) and previous injury profiles. Other factors which can be considered are injury aetiology models and population demographics which combine to provide an accurate picture of the risk, frequency and type of injuries. These data are gathered through the injury surveillance system by either paper or electronic media [6,7].

Injury surveillance systems have been implemented at various levels of sports participation. Systems have been established and implemented at the college level [8] and high school level [9] and in professional sports [10,11]; however, the practice is not universal across all sports and there are still knowledge gaps to be addressed in different contexts and sports. Professional sporting bodies have been widely acknowledged to be at the forefront of injury surveillance since the USA-based National Football League (NFL) established its surveillance system in 1980 [12]. Athlete monitoring at the professional level has direct financial implications for sports clubs as it is associated with managing and possibly prolonging the careers of highly talented individuals.

Whereas the focus of participation at the amateur and recreational level is primarily associated with various health implications for the broader population, such as motor skill development, socialisation, teamwork, stress management and general health benefits [13,14]. However, both amateur and recreational activity are a leading cause of injury in adolescents, accounting for >30% of injuries in this population across many countries, with the highest rates of injury for males and females reported in amateur team sports [15].

Systems such as the National Electronic Injury Surveillance System (NEISS) may provide data on serious sports injuries through emergency department records. There is a need to capture data about less traumatic and minor injuries that may not require a visit to a hospital. Professional or elite sports usually have a certified athletic trainer, allied health professional or other medical staff employed to record the injury data to a surveillance system [10,12]. However, in amateur and recreational sports, injury surveillance systems rely on volunteers, coaches or self-reporting by the athletes participating. As has been established in previous research, there are large discrepancies in the reliability of injury rate reporting from volunteers and athletes [6]. The World Health Organisation (WHO) has recognised that involving medical staff or trained designates will support a more consistent injury surveillance, but this may not always be possible in amateur and recreational sports activity. With more systems in the amateur domain, the challenge is to ensure a consistency of reporting of injury surveillance data [16].

As technology advances and hardware becomes more accessible, methods of data collection and data storage will evolve. With the widespread availability and use of mobile devices, it is possible to almost immediately record injury data to a central data storage system. As well as the mechanism, location and classification of the injury, available sensors and devices such as global positioning systems (GPSs) or global navigation satellite systems (GNSSs) can provide external data such as speed and force of collisions. The aim of this review was to identify systems used for ongoing injury surveillance within sports clubs and organisations at both the professional and amateur level. This study updates the review of injury surveillance systems conducted by Ekegren et al., (2015) [17]. Suitable injury surveillance requires a standard system for classifying injuries, together with systems for keeping records on individual cases and producing summary statistics [2]. A standard classification, like the Orchard Sports Injury Classification System (OSICS), makes comparison of data from multiple systems more straightforward.

The methodologies of those systems identified are presented to provide useful comparisons between them and to help in the design and implementation of systems in new settings and contexts.

## 2. Objectives

### 2.1. Literature Search

A structured search of the literature was performed to identify peer-reviewed articles and grey literature on ongoing sports injury surveillance systems. The timeframe focused on research published between 2015–2022, as the most recent injury surveillance systems review was published by Ekegren et al. in 2015 [17]. Systematic searches of MEDLINE, CINAHL, EMBASE, SCOPUS and ProQuest were carried out to identify relevant publications. The search strategy for MEDLINE is presented in Figure 1. The searches of the other respective databases were carried out in a similar manner.

Using the same search terms that Ekegren used, the titles, abstracts and keywords were searched and results were limited to the English language. Google (Google LLC, Mountain View, CA, USA) searches were conducted to locate grey literature on identified injury surveillance systems with no associated peer-reviewed publications, using the name of the system as the search term.

### 2.2. Study Selection Criteria

The inclusion criteria were developed to document systems that are in ongoing use for athletes or sport participants in clubs or organisations. All publication types, reports, theses and peer-reviewed journals were included in the review.

Sport from all levels, from professional down to recreational, were included with equal consideration for both male and female athletes of all ages. As this paper is focused on ongoing sports injury surveillance systems, any systems using mortality, hospital, clinic or insurance data were excluded.

Papers that did not present methodological data for the injury surveillance system were excluded as they would not add to the breakdown of the data stored by the system, the injury definition for that system or the person responsible for recording the information.

Studies that presented systems that were no longer in use were not included in this review. Surveillance is the “ongoing collection of health information” [2], so systems that had ceased operation or those that were not intended for continuous operation were not of interest.

Titles and/or abstracts were reviewed in the first instance, with any not meeting inclusion criteria excluded. If there was any doubt, then the full article was assessed. Only the most recent publications or the one providing the most details were included in the event of multiple publications for a single system.

### 2.3. Data Synthesis

Details of the surveillance systems were extracted from the papers during full review. These included the name of the system, the country of origin, the period over which it was used, the population under study, the level of the sport, the injury definition and the injury classification, where available.

While the systems could potentially store data for athletes and sports at all levels, they have been classified in this review based on the reported level of competition where they were used or the professional status of the athletes under surveillance in the studies referenced. For example, the International Olympic Council injury and illness surveillance system for multi-sport events is used for elite athletes and the New Jersey Safe Schools Program reports on amateur athletes.

## 3. Results

From the original 1198 papers returned through database searches, a total of 109 full-text articles were reviewed, with 50 of these being excluded. Thirty-five of these described systems that were not ongoing surveillance, three were no longer in use, nine were topic commentaries and three were consensus statements. Figure 2 shows the results of the searches.

Twenty-one systems are listed, and their methods expanded, in Table 1. These were identified through searches and subsequent review of papers and are included in this review:Welsh Rugby Union Injury Surveillance Program (WRU-ISP) [10]Canadian Intercollegiate sport Injury Registry (CISIR) [4,18,19,20]Paralympic Injury and Illness Surveillance System (WEB-IISS) [21,22,23]National Collegiate Athletic Association Injury Surveillance System (NCAA-ISS)—USA [8,9,24,25]National Athletic Treatment, Injury and Outcomes Network (NATION)—USA [25,26]High School Reporting Information Online Sports Injury Surveillance System (HS-RIO)—USA [8,9,27]National Football League Injury Surveillance System—USA(NFL-ISS) [12]The International Olympic Council injury and illness surveillance for multi-sport events [28,29,30,31,32,33,34,35,36,37,38]National Interscholastic Cycling Association Injury Surveillance System (NICA-ISS)—USA [6,16]American Ultimate Disc League Injury Surveillance Program (AUDL-ISP) [39]Football Australia Injury Surveillance spreadsheet [40]Athlete Management System—Australia [41,42,43,44]Aspetar Injury and Illness Surveillance Program—Qatar [11,45]Major League Baseball Health and Injury Tracking System (MLB-HITS) –USA [46]Athlete Monitoring System—Ireland [47]Cricket Australia Athlete Management System [48]National Gaelic Athletic Association Injury Surveillance Database—Ireland [49,50]SAP Sports One—Germany [51]New Jersey Safe Schools Program School-Based Young Worker Incident Surveillance System (NJ Safe Schools)—USA [52]Irish Rugby Injury Surveillance Web-Based System (IRISweb) [53]World Rugby Injury Surveillance System [54]

Twelve of the systems record data on elite or professional athletes [4,10,11,12,21,22,23,28,29,30,31,32,33,34,35,36,37,38,39,40,42,43,44,45,46,47,48,49,50,51,54]. Seven of the systems record data on amateur or community athletes [6,8,9,16,18,19,24,25,26,27,41,52,53] and two of the systems record for both amateur and elite athletes [4,41]. Eleven of the surveillance systems record data within various football codes (American Football: *n* = 4, Rugby: *n* = 3, Soccer: *n* = 3, Gaelic: *n* = 1) [10,11,12,25,38,40,42,45,49,50,51,53,54]. Seven of the systems record data within multiple sport disciplines [8,9,21,22,23,24,26,28,29,30,31,32,33,34,35,36,37,41,43,44,52]. Two of the systems record data for hockey (Ice Hockey: *n* = 1, Field Hockey; *n* = 1) [4,18,19,20,47] with one each for baseball [46], mountain biking [6,16], basketball [9], cricket [48] and ultimate frisbee [39]. Nine of the systems collect data on male athletes only [10,11,12,45,46,47,49,50,51,54] with the remaining 12 collecting data on both male and female athletes [4,6,16,18,19,20,21,22,26,28,29,30,31,32,33,34,35,36,37,38,39,42,43,44,48,52,53]. There are no systems dedicated to collecting female data only. There are still no systems aimed specifically at recording injury data in children. Eight of the systems are US-based [6,12,16,24,26,27,39,46,52], six are based in Europe [10,21,22,23,47,49,50,51,53,54], three in Australia [40,41,42,43,44,48], one in Canada [4,18,19,20], one in Qatar [11,45] and two with no unique national affiliations [28,29,30,31,32,33,34,35,36,37,38,54]. Sixteen of the systems provide for consensus sampling [4,6,8,9,10,11,12,16,18,19,20,21,22,23,24,25,26,27,28,29,30,31,32,33,34,35,36,37,38,39,40,42,43,45,46,47,48,53,54] and five of the systems record data on a convenience sample of their target data population [41,44,49,50,51,52]. The systems listed here have been operating between four and 42 years—the National Football League (NFL) system, established in 1980, is still the longest running, while the NICA-ISS went live in 2018. Seven of the studies [55,56,57,58,59,60,61] used standardised forms like the Oslo Trauma Research Center questionnaires as the basis of their work and, on further examination, had not created a surveillance system that could be described like those listed.

Table 1 provides the methods of the surveillance systems identified. One of the systems does not require a medical diagnosis to make a reportable injury. Although data about medical attention is recorded, the need for medical attention is a binary variable determined by the athlete themselves [39]. Four systems record an injury with a medical diagnosis regardless of any time lost [21,22,23,28,29,30,31,32,33,34,35,36,37,38,47,48]. The remaining systems utilise both a medical diagnosis and time loss definition for recordable injuries. One of the systems allows for the athlete to enter the injury data, although this is confirmed by medical staff before final saving [47]. One system provided for trained designated personnel to enter the injury data [6,16], while the other systems have data entered by certified athletic trainers, physiotherapists or medical staff associated with the teams.

## 4. Level of Evidence

The objective of this review was to identify injury surveillance systems which were in use up to 2022 following on from Ekegren et al.’s study in 2015 [17]. Ekegren et al. identified fifteen systems, with eleven of those (73%) in use for professional- or elite-level sports. This updated review has identified twenty-one injury surveillance systems, with fourteen of those systems (67%) being used in professional- or elite-level sports. Some of the new systems (since 2022) include the Welsh Rugby Union Injury Surveillance Program (WRU-ISP), the Paralympic Injury and Illness Surveillance System (WEB-ISS), the National Interscholastic Cycling Association Injury Surveillance System (NICA-ISS), the National GAA Injury Surveillance Database and the Irish Rugby Surveillance Web-Based System (IRISweb).

### 4.1. Resource Requirements

From the systems reported, twelve operate within various football codes [10,11,12,25,38,40,42,45,49,50,51,53,54] and nine systems record data for male athletes only [10,11,12,45,46,47,49,50,51,54]. Two of the systems record data for the Olympics and Paralympics—both catering for a global field of athletes with a global audience. This dominance of surveillance systems in the professional, male and global sporting arena reflects the need for the financial and human resourcing as identified by the WHO [2]. The predominant reason for funding being central is due to the time-consuming process of assembling and maintaining a database and registry [62].

Many organisations may have challenges in sourcing funding for the specific purpose of injury surveillance. A significant factor for effective injury surveillance in various sports is the availability of funding from their respective national and international sports organisation [63]. Most systems operate in professional sport which can support the necessary resources of dedicated medical teams, allied health professionals and professional coaches.

### 4.2. Common Systems

With adequate funding and support, the longevity of an injury surveillance system has been successfully demonstrated at both the professional and amateur level. Six of the systems identified in both this review and Ekegren’s review of 2015 have provided data for injury prevention studies and research for a substantial time. The NCAA-ISS (1982), HS-RIO (2005), NFL-ISS (1980), MLB-HITS (2010) in the USA, the Cricket Australia Athlete Management System (AMS pre-2000) and the IOC injury and illness surveillance for multi-sport events (2008), which is widely implemented, have all been in use for over a decade, with the NFL and the NCAA systems dating from 1980 and 1982, respectively. All these systems are fully supported both medically and financially, with participation by the athletes being compulsory within their disciplines and contexts.

The NFL-ISS, MLB-ISS, Cricket Australia AMS and the IOC system all cater for professional athletes, while the NCAA-ISS and HS-RIO gather data on amateur athletes. These systems all have a compulsory participation and support from governing bodies and professional leagues.

Since 2015, twelve systems have been reported in the literature that did not feature in Ekegren’s 2015 review. Although some of these were operating from 2007, the published data were not available until post-2015 [6,10,11,16,21,22,23,26,39,40,45,47,49,50,51,52,53]. These include systems in rugby, GAA, soccer, Paralympics, cycling and ultimate frisbee. Some of these were developed following research carried out by athletes within the discipline. More recent systems are more aligned in injury definition but self-reporting by the athlete is still not considered as optimal. Several of the studies were dealing with concussion and head injuries that require clinical assessment to properly diagnose. It is not possible for an athlete to self-report in this case.

### 4.3. Athlete Level

There has been an emergence of systems operating at the amateur or community level of participation. This review identified nine systems in use for amateur athletes, compared to four systems in Ekegren’s 2015 review. These include the National GAA Injury Surveillance Database, NJ Safe Schools and IRISWeb. This indicates that more research is being carried out at the amateur level in sports globally, with eight systems reported active since Ekegren’s survey being based outside the US.

There are many factors that could contribute to this increase, including cheaper, smaller and more reliable technology and a larger population to study due to an increase in participation at the recreational and amateur levels [64]. The role of physical exercise in physical and mental wellbeing is widely understood [13,14], but as participation increases, so too does the risk and occurrence of injury. This increase in participation is generally at the amateur level, as communities become more active through recreational engagement, so the research is opening to this demographic.

### 4.4. Data Entry and Collection

Data entry for all but one of the identified systems, namely the Athlete Monitoring System for Field Hockey (AMS), is completed by medical staff associated with the team, regardless of level of competition. These can be athletic trainers, physiotherapists, medical doctors or nurses. Some of the systems have data recorded by trained designated reporters, alternate injury recorders and, in only one case, the players. In the AMS, the entries are still reviewed by qualified personnel to confirm the injury definition criteria. The Injury Surveillance Guidelines of the WHO state that completing an injury report requires a high level of literacy and an understanding of the mechanism of injury that is generally beyond the knowledge of the average layperson. The involvement of medical staff or trained designates in all the systems will support more consistent injury surveillance.

With a move to an online world and connected devices, web-based or electronic data collection is available in all the systems identified, although paper alternatives are available in some. Consistent and valid injury surveillance practices allow for the comparison of injury burden from season to season and can determine the effectiveness of an injury prevention intervention [65]. An electronic form, complete with drop-down lists and checkboxes, minimises the amount of free text input and can aid this more consistent recording of data elements. The use of the OSICS, the Sports Medicine Diagnostic Coding System (SMDCS) or similar when recording the data will encourage consistent terminology in injury reporting.

### 4.5. Injury Definition

The injury definition for each of the systems identified is presented in Table 1. Standardised definitions are needed when different sports activities are to be compared or when data from several different sources about a given sport are to be collated [66]. While there are some variations in the injury definitions, most incorporate a time-loss element. This is the most commonly used definition, particularly in longer-term surveillance programmes in team sports [67]. This is a narrow definition of injury and one that is likely to cause an under-reporting of incidents. The criteria for time loss varies from an inability to complete the session (in which injury occurred) to preventing the player taking full part in activity where the injury has been there for a period of >24 h from midnight at the end of the day that the injury has occurred [4,16].

Most of the systems couple this time loss with a medical attention component for injury recording. Without medical support in amateur sports, the use of a medical-attention definition can cause problems, and even within a professional team-sport environment, there are likely to be differences in interpretation of what constitutes medical attention [67]. All the systems described here have associated medical personnel for the recording of injuries, even when the athletes are entering the data initially.

The WEB-ISS, IOC ISS for multi-sport events, AMS and the Cricket Australia Athlete Management systems record the injury regardless of time lost by the athlete. Two of these—WEB-ISS and the IOC-ISS—are used in events (Olympics and Paralympics) where polyclinics are available and athletes have relatively uniform access to medical care [67]. Basing a recordable injury purely on medical attention is appropriate for this type of event. However, in tournaments which do not provide such services but rely purely on team physicians for treatment and data collection, the use of a medical-attention definition may lead to a systematic under-reporting from athletes and teams with less intensive medical coverage [67]. The NICA-ISS, NATION-IS and the NCAA-ISP all incorporate the term injury event to collect and quantify injury data. The injury event can be used to capture multiple injuries that may occur in a single event for the athlete. 

The alignment of injury definition among the papers reviewed is in line with the specific IOC goal to further encourage consistency in data collection, injury definition and research reporting [1]. This resulted from an IOC expert panel meeting in 2019 to update recommendations for the field of sports epidemiology.

Systems are discipline or tournament specific and do not consider multi-sport athletes and the possibility of previous injury in a different sporting event. Consensus statements have been published in several sports (rugby, cricket and soccer). These consensus statements provide detailed approaches for injury surveillance studies within specific sports, but they may not be appropriate where several diverse sports are being compared [7].

### 4.6. Limitations

Despite the systematic search method, it is still possible that some injury surveillance systems have not been included.

Some injury surveillance systems have been used in studies that report data for male participants only. Consequently, they may have been listed with a “Population Under Surveillance” as male only, but it is possible that they have been used, or can be used, for female participants as well. There may also be systems currently in development, testing or early stages of use with no associated published literature. Studies that used systems no longer in use were not included. These include some systems that used widely available electronic surveys to gather the data coupled with Microsoft Excel (v.365) for storage [68,69,70,71].

## 5. Conclusions

Injury surveillance systems are still used widely within research and the sporting arena. Professional sports have greater resource availability and access to medical professionals for long-term surveillance and data collection. Although still predominantly applied in the professional landscape, there is an increasing interest in the applicability and effectiveness of injury surveillance in amateur sport since the last review in 2015. The evidence of successful injury prevention programmes driven by the surveillance data in the professional domain may be a catalyst for this development. As happens in may sports, the practices which occur in professional sports may often trickle into amateur sports over time. However, consistent medical resources are required for a thorough implementation of injury surveillance systems, and these are more readily available in the professional sports when compared to amateur settings.

There is a dearth of published research concerning injury surveillance for multi-sport athletes in amateur sports. While systems cater for multiple sports and multiple injuries, amateur multi-sport athletes are not uncommon, especially at the juvenile level, where they have not yet specialised.

Injury definitions are more aligned with the consensus statements. The newer systems all use these as the basis for definition, regardless of the mechanism of injury. Despite the time investment needed for an injury surveillance system, the increase in the number of amateur focused systems must be seen as a positive development in the monitoring of sports injuries and the development of injury prevention systems.

## Figures and Tables

**Figure 1 jfmk-09-00177-f001:**
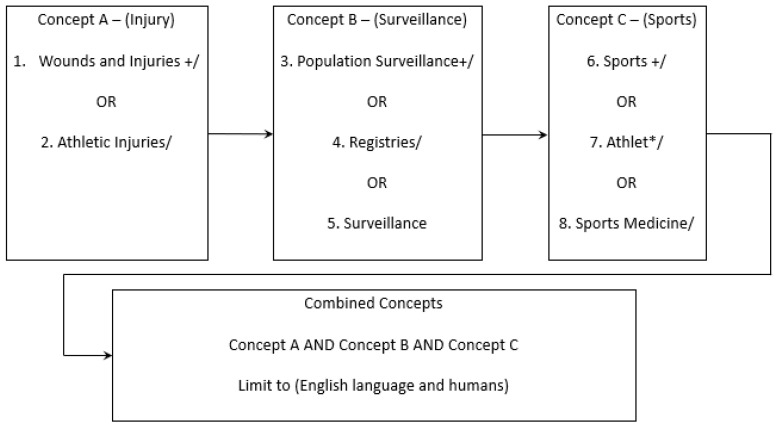
MEDLINE search strategy, consisting of medical subject headings and free text terms (using ‘*’) searched for in titles, abstracts and keywords.

**Figure 2 jfmk-09-00177-f002:**
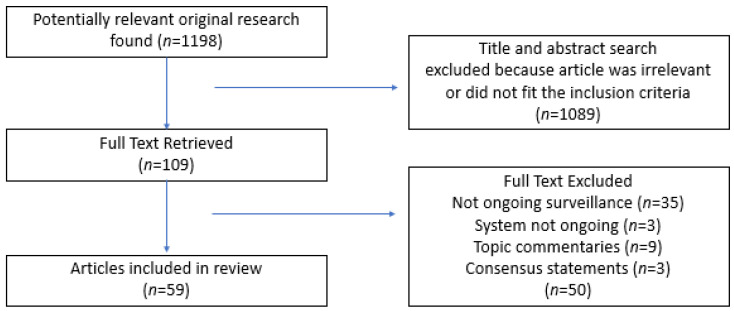
Search results’ retrieval and review data.

**Table 1 jfmk-09-00177-t001:** Methodological details of injury surveillance systems identified.

No.	Name	Organisation	Sport/Level	Country of Origin	Population under Surveillance	Coverage	Years of Operation	Injury Definition	Who Records Data?	How Is Data Recorded?
1	WRU-ISP	Welsh Rugby Union	Professional rugby	Wales	Male professional Rugby Union players	All professional clubs in Wales	From 2012	Time loss (Any physical complaint sustained by a player during the season that rendered the player unavailable for match selection for more than one day).	Medical staff	Electronic injury database
2	CISIR		Youth ice hockey	Canada	Male + female amateur and elite athletes	Convenience samples of players at different age and levels	From 1998	Time loss + medical attention (injury requiring medical attention, the inability to complete the session and/or time loss from activity).	Medical staff or a trained designate	Electronic injury database
3	WEB-IISS		Multiple sports	UK	Male + female paralympic athletes	Convenience samples at Paralympic Games	From 2012	Medical attention (an injury that required medical attention, regardless of the consequences with respect to absence from competition or training)	Medical staff	Web-based system
4	NCAA-ISP	NCAA	Multiple sports	USA	Male + female university athletes	Convenience samples of university athletes	From 1982	Medical attention (time loss (TL) or non–TL, that occurred as a result of participation in an organized practice or game and required attention from an AT or physician)	Certified athletic trainer	Electronic injury database
5	NATION	Datalys	Multiple sports	USA	High school athletes	Convenience samples of high school athletes	From 2011	Medical attention (time loss and non-time loss, occurred in sanctioned event and required attention).	Athletic trainers	Web-based system
6	HS-RIO	Datalys	Multiple sports	USA	High school athletes	Convenience samples of high school athletes	From 2005	Medical attention (time loss and non-time loss, occurred in sanctioned event and required attention).	Athletic trainers	Web-based system
7	NFL-ISS	NFL	Professional American football	USA	Male, professional American football players	All NFL teams	From 1980	Time loss (any injury causing the athlete to miss at least one day of participation in either practice or games).	Certified athletic trainer	Web-based system
8	IOC ISS for Multi-sport Events	IOC	Multiple sports	Multiple Countries	Male + female elite athletes	Athletes at Olympic and World Championship events	From 2008	Medical attention (any injury newly occurred, or other medical conditions due to competition and/or training that required medical attention regardless of consequences for absences.	Medical staff	Paper-based or electronic forms
9	NICA-ISS	NICA	Cycling, mountain biking	USA	Male + female athletes	All student/youth cyclists	Developed in 2016 Live in 2018	Time loss + medical attention. Injury event—physical event occurring to a single rider during sanctioned event that requires medical attention or misses time beyond the day of injury.	Designated reporter	Web-based system
10	AUDL-ISP	AUDL	Professional frisbee	USA	Male professional athletes	Convenience sample of 16 all male teams included.	From 2016	Time loss (physical harm that happened while the player was participating in an AUDL competition or practice and caused the player to miss part of a competition or practice).	Athletic trainer + team representative	Web-based system
11	Football Australia IS spreadsheet	Football Australia	Professional soccer	Australia	Male professional athletes	A-league teams (*n* = 10)	From 2012 to 2018	Time loss + medical attention (any physical complaint that required medical attention resulting in a missed A-League match).	Team physio	Electronic injury database
12	Athlete Management System	Smartbase, Fusion Sport	Multiple sports	Australia	Male + female amateur + professional athletes	Convenience sample from athletes	From 2003	Time loss + medical attention (any recorded medical attention sustained during training or competition that results in an athlete being unable to participate for more than one day).	Medical staff	Electronic injury database
13	Aspetar IISP	Aspetar	Soccer	Qatar	Professional soccer players QSL	First division clubs (*n* = 14 of possible 18)	From 2013	Time loss (any physical complaint resulting from playing soccer and preventing full participation in future training or match).	Club doctor	Electronic injury database
14	MLB—HITS	MLB	Baseball	USA	Male professional baseball players	All teams in major and minor leagues	From 2010	Time loss (any physical complaint sustained by a player that affects or limits participation in any aspect of baseball related activity).	Certified athletic trainer + team doctor	Electronic medical record
15	Athlete Monitoring System	Athlete Monitoring	Field hockey	Australia	Male elite hockey players	All adult male hockey teams in IHL	Unknown	Any physical complaint sustained from a field hockey match or training session, irrespective of the need for medical attention or time loss from activities.	Players, but confirmed by medical staff	Electronic injury database
16	Cricket Australia Athlete Management System	Cricket Australia	Cricket	Australia	Male + female amateur + professional players	Convenience sample from players	Pre 2000	Medical diagnosis for head impact and concussion regardless of time loss.	Medical staff	Web-based system
17	National GAA Injury Surveillance Database	GAA	Gaelic football	Ireland	Elite male players	15 male football teams	From 2007	Time loss + medical diagnosis (any injury that prevents a player taking a full part in training and match play planned for that day, where the injury has been there for a period of >24 h from midnight at the end of the day that the injury was sustained)	Medical staff	Web-based system
18	SAP Sports One	SAP,	Soccer	Germany	Elite youth male players	Convenience sample of male players.	From 2015	Time loss + medical diagnosis (any physical complaint that restricted full participation in soccer training or matches)	Medical staff	Web-based system
19	NJ Safe Schools		Multiple sports	USA	Amateur male + female athletes	Convenience sample from athletes	From 2013	Medical attention (any primary or repeated concussion-type injury in PE, sports and athletics practice or competitive events)	Certified athletic trainers + school nurses	Web-based system
20	IRISweb	IRFU	Rugby	Ireland	Amateur male and female players	Convenience sample from clubs	From 2017	Medical attention injury, time loss injury (any physical complaint sustained during training or match).	Medical staff or alternate injury recorder	Web-based system
21	World Rugby ISS	WRU	Rugby	Multiple countries	Male professional players	Convenience sample from players	From 2007	Time loss (Any physical complaint sustained during a RWC match/training session preventing player from taking a full part in all training activities or match play for more than 1 day following the day of injury, irrespective of whether match or training sessions were actually scheduled.	Medical staff	Unknown

## Data Availability

Data are available on request from the corresponding author.

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
