# Peer review of "Sports Injury Surveillance Systems: A Scoping Review of Practice and Methodologies"

_jfmk, 2024, doi:10.3390/jfmk9040177_

Round 1
Reviewer 1 Report
Comments and Suggestions for Authors
Dear Authors,
Thank you for your interesting, up-to-date meta study on approaches for monitoring injury incidence and prevalence in athletic populations.
In general, your study provides a valuable current summary of the current situation worldwide.
However, there are some issues that you should carefully address before I can recommend the publication of your article in JFMK:
Major issues:
· Line 104: Why did you not include PubMed as a resource for your survey? Please provide a reasonable rationale for this choice, or add PubMed results to your research.
Minor issues:
· Lines 175 through 187: Please provide the countries where the systems have been set up or used already here.
· There seem to be many double spaces in your text. Please remove them all.
· Please decide on consistent BE or AE spelling. Currently, you use both, e.g. “programme” an “program”.
· Please add the country when you mention the NFL etc. the first time, even though it may seem obvious to you.
· Line 80: Please be precise in your technical term GPS: Do you really refer only to GPS systems, or merely GNSS systems?
· Line 106: what is a “r database”?
· Line 382 Please replace “don’t” by “do not”.
Best regards
Your reviewer
Comments on the Quality of English LanguageThere are just a few spelling inconsistencies that can be easily remedied.
Author Response
NOTE TO REVIEWER 1
Thank you for your time in reviewing this article in the detail that you have and for the comments you have returned. I have read your comments carefully and address each in turn below with reference to the line numbers within the paper.
Major issues:
- Line 104: Why did you not include PubMed as a resource for your survey? Please provide a reasonable rationale for this choice or add PubMed results to your research.
RESPONSE
The purpose of this article was to replicate and update the searches of Christina Ekegren in her paper “Sports Injury Surveillance Systems: A Review of Methods and Data Quality” published in 2015. Part of this was to reference the same databases that were used then and update them from 2015. As PubMed was not included in that paper, it was not searched in this review.
Minor issues:
- Lines 175 through 187: Please provide the countries where the systems have been set up or used already here.
RESPONSE
This list has been updated to include the country names for reader clarity. The International Olympic Council, the World Rugby Injury Surveillance System and others with the country name in the title have been left as they are. For example, the Cricket Australia Athlete Management System.
- There seem to be many double spaces in your text. Please remove them all.
RESPONSE
130 instances of double spaces have been removed.
- Please decide on consistent BE or AE spelling. Currently, you use both, e.g. “programme” an “program”.
RESPONSE
Using BE spelling with the exception of system names and references.
- Changed two occurrences in text on line 38 and one on line 43.
- Systems names on line 171, 185, 189, 197 and 266 have been left unchanged.
- Reference 8 and 26 remain unchanged.
- Please add the country when you mention the NFL etc. the first time, even though it may seem obvious to you.
RESPONSE
NFL updated on line 54 to indicate it is USA based.
This is the only reference to a particular system before the list presented at line 161. A brief note for the country where the system is in use has been added for the list between lines 171 and 200 to clarify for the reader.
- Line 80: Please be precise in your technical term GPS: Do you really refer only to GPS systems, or merely GNSS systems?
RESPONSE
Have edited the line (now line 81) to include reference to both systems as…
“As well as the mechanism, location and classification of the injury, available sensors and devices such as global positioning systems (GPS) or global navigation satellite systems (GNSS) can provide external data such as speed and force of collisions.”
- Line 106: what is a “r database”?
RESPONSE
The “r” is a typo in this case and the correct sentence is “The searches of the other respective databases were carried out in a similar manner.”. This has now been corrected in the paper.
- Line 382 Please replace “don’t” by “do not”.
RESPONSE
This has been corrected in the paper. No other abbreviations of this nature were found in the paper.
Reviewer 2 Report
Comments and Suggestions for Authors
This manuscript provides a comprehensive overview of sports injury surveillance systems, covering a range of systems from professional to amateur levels. The scope and purpose of the study are clearly stated in the abstract, offering readers a macro perspective on current injury surveillance practices. However, there are several critical issues that need to be addressed to enhance the clarity, accuracy, and overall quality of the manuscript.
1.LINE189 claims that a total of 21 systems were documented, with 12 systems recording data on elite or professional athletes and 7 systems recording data on amateur or community athletes. However, these numbers add up to 19, not 21. The authors need to carefully re-examine the classification and accounting of these systems. There may be missing or misclassified systems that need to be correctly identified and included.The abstract mentions that "two-thirds of the systems are exclusively in the professional/elite landscapes." However, according to the main text, only 12 out of the 21 systems are classified as elite or professional, which represents approximately 57%, not 66.7% (two-thirds). The authors must either revise the abstract to reflect the correct percentage or provide clarification if there are additional systems that should be included in the elite/professional category.
2.The manuscript lacks a clear explanation of the criteria used to distinguish between "elite/professional" and "amateur" systems. The authors should provide detailed criteria that define these classifications. For example, are these based on the level of competition, the professional status of the athletes, or the context in which the systems are used? A clear definition is essential for the validity of the study's conclusions.
3.On line 192, it is stated that "Twelve of the surveillance systems record data within various football codes (American Football: n=4, Rugby: n=4, Soccer: n=2, Gaelic: n=1)." However, these numbers total 11, not 12 as mentioned. The authors should ensure that the correct number of systems is accurately reflected in the manuscript.
4.The manuscript states that 62 articles were included in the review, but the reference list does not appear to contain 62 citations. The authors need to verify that all the included studies are properly cited in the reference list. If some studies were missed, they should be added. If multiple studies are combined under single references, this should be clearly explained.
5.The authors should carefully review the reference list to ensure consistency in citation style, including the correct use of punctuation, italics, and abbreviations as required by the journal. For instance, references 8 and 9 clearly belong to the same journal, yet their formats differ, which indicates a need for consistent formatting throughout the reference list.
6.The final column in Table 1, labeled "Years of operation," is incomplete.The authors should ensure that this column is fully completed for all systems listed, or if the information is unavailable, they should explicitly state this.
7. The manuscript would benefit from the inclusion of clear and informative charts and visualizations to better illustrate the key findings and comparisons.
Author Response
NOTE TO REVIEWER 2
Thank you for your time reviewing this paper in detail and the salient points that you have raised below through this. We have read your comments carefully and attempted to answer each in turn below.
This manuscript provides a comprehensive overview of sports injury surveillance systems, covering a range of systems from professional to amateur levels. The scope and purpose of the study are clearly stated in the abstract, offering readers a macro perspective on current injury surveillance practices. However, there are several critical issues that need to be addressed to enhance the clarity, accuracy, and overall quality of the manuscript.
1.LINE189 claims that a total of 21 systems were documented, with 12 systems recording data on elite or professional athletes and 7 systems recording data on amateur or community athletes. However, these numbers add up to 19, not 21. The authors need to carefully re-examine the classification and accounting of these systems. There may be missing or misclassified systems that need to be correctly identified and included.The abstract mentions that "two-thirds of the systems are exclusively in the professional/elite landscapes." However, according to the main text, only 12 out of the 21 systems are classified as elite or professional, which represents approximately 57%, not 66.7% (two-thirds). The authors must either revise the abstract to reflect the correct percentage or provide clarification if there are additional systems that should be included in the elite/professional category.
RESPONSE
Sincere thanks for the comment, the reviewer is correct on this statement. However, the last part of the sentence on line 189 (now 202) is that two of the systems record data on both amateur and elite. Adding these two brings the total number of systems to 21.
The two-thirds used here is a rounding that would be better replaced with the percentage number for better accuracy (as suggested) in the Abstract. The revised sentence is as follows:-
A total of 21 systems were documented as being used in ongoing injury surveillance with 57% of these exclusively in the professional/elite landscapes and 33% at the amateur level.
2.The manuscript lacks a clear explanation of the criteria used to distinguish between "elite/professional" and "amateur" systems. The authors should provide detailed criteria that define these classifications. For example, are these based on the level of competition, the professional status of the athletes, or the context in which the systems are used? A clear definition is essential for the validity of the study's conclusions.
RESPONSE
Thanks for this observation; the systems described here were classified by the level of competition and the professional status of the athlete as reflected in the referenced study. The systems themselves could technically be used in several domains so the classification is based on the reported information from the studies. This has been updated in the document under the section 2.3 Data Synthesis (line 147) with the addition of the following paragraph.
While the systems could potentially store data for athletes and sports at all levels, they have been classified in this review based on the reported level of competition where they were used or the professional status of the athletes under surveillance in the studies referenced. For example, the International Olympic Council Injury and Illness Surveillance System for Multisport Events is used for elite athletes and the New Jersey Safe Schools Program reports on amateur athletes.
3.On line 192, it is stated that "Twelve of the surveillance systems record data within various football codes (American Football: n=4, Rugby: n=4, Soccer: n=2, Gaelic: n=1)." However, these numbers total 11, not 12 as mentioned. The authors should ensure that the correct number of systems is accurately reflected in the manuscript.
RESPONSE
Thanks for this clarifying point; the football codes represented should be 11 and this has been corrected in the text. Thanks to the reviewer, an error has also surfaced in that the correct numbers for Rugby and Soccer are both 3, rather than the 4 and 2 here. These have now been stated correctly in the text of the paper.
4.The manuscript states that 62 articles were included in the review, but the reference list does not appear to contain 62 citations. The authors need to verify that all the included studies are properly cited in the reference list. If some studies were missed, they should be added. If multiple studies are combined under single references, this should be clearly explained.
RESPONSE
Thank you to the reviewer for drawing attention to the references included. The correct references are now included. All references have now been included with additional text on line 223 to indicate that these had been questionnaires used for surveillance but the details of the storage of the data is not available to allow for description in the same way as the listed systems.
An updated figure for the search results has been added on line 162. There were 2 errors in the diagram for the articles excluded which was confirmed when reviewing the references. The appropriate text has been corrected in the Results section as well.
5.The authors should carefully review the reference list to ensure consistency in citation style, including the correct use of punctuation, italics, and abbreviations as required by the journal. For instance, references 8 and 9 clearly belong to the same journal, yet their formats differ, which indicates a need for consistent formatting throughout the reference list.
RESPONSE
This results from one of the issues when using Endnote. The references have now been checked in the final version for consistency of formatting. References 2, 7, 9, 11, 18, 27, 55, 67 have been updated for a more consistent format.
6.The final column in Table 1, labeled "Years of operation," is incomplete. The authors should ensure that this column is fully completed for all systems listed, or if the information is unavailable, they should explicitly state this.
RESPONSE
The table has now been updated with either the years or the word “Unknown” if the years could not be established.
- The manuscript would benefit from the inclusion of clear and informative charts and visualizations to better illustrate the key findings and comparisons.
RESPONSE
It was felt that the tabular representation for data presentation in case was the most appropriate in this instance by the research team.
Round 2
Reviewer 2 Report
Comments and Suggestions for Authors
Thank you to the authors for their hard work. However, the use of revision mode has affected the readability of the figures and tables for the reviewers. I recommend paying closer attention to this in future revisions.